# Comparison between Fine Needle Aspiration Cytology with Histopathology in the Diagnosis of Thyroid Nodules

**DOI:** 10.3390/jpm13081197

**Published:** 2023-07-28

**Authors:** Michael Osseis, Georges Jammal, Daniel Kazan, Roger Noun

**Affiliations:** Department of General Surgery Hôtel-Dieu de France Hospital, Saint Joseph University of Beirut, Beirut 1107 2180, Lebanon; georgejammal17@gmail.com (G.J.); danielkazan94@gmail.com (D.K.); rogernoun@gmail.com (R.N.)

**Keywords:** thyroid nodule, ultrasound-guided fine needle aspiration biopsy, diagnosis, histopathology

## Abstract

Background: Accurate diagnosis of thyroid nodules is crucial for avoiding unnecessary surgeries and enabling timely treatment. Fine needle aspiration cytology (FNAC) and ultrasound are commonly employed diagnostic techniques, but their reliability is debated. This study aimed to compare the diagnostic accuracy of FNAC and ultrasounds using histopathology as the reference standard. Methods: A retrospective review was conducted on 344 patients who underwent thyroidectomy between January 2017 and May 2022. An ultrasound and FNAC were performed before surgery, and histopathological findings were compared. Statistical analyses were conducted to calculate sensitivity, specificity, positive predictive value (PPV), negative predictive value (NPV), false positive rate, false negative rate, and overall accuracy for each diagnostic method. Results: Among the study population, 38.67% of thyroid tumors were malignant. Ultrasound showed a sensitivity of 68.18%, specificity of 76.55%, PPV of 64.74%, NPV of 79.20%, and overall accuracy of 73.31%. FNAC had a sensitivity of 89.31%, specificity of 48.44%, PPV of 78%, NPV of 68.89%, and accuracy of 75.89%. The correlation coefficient between ultrasound and FNAC was 0.512 (*p* < 0.0000001). Ultrasound correlated with histopathology with a coefficient of 0.408 (*p* < 0.0000001), while FNAC with histopathology had a coefficient of 0.304 (*p* < 0.00001). The correlation coefficient between these three diagnostic methods was 0.423 (*p* < 0.0001). Conclusion: In the diagnosis of thyroid nodules, both FNAC and ultrasound demonstrated moderate diagnostic accuracy. Ultrasound showed a higher specificity, while FNAC exhibited a higher sensitivity. Combining these techniques may improve diagnostic accuracy. Further research and the development of more reliable diagnostic methods are warranted to optimize the management of thyroid nodules and avoid unnecessary surgeries.

## 1. Introduction

Thyroid nodules are frequently detected among adults, with a prevalence rate of 40% in individuals aged 60 and above. Among these nodules, only a small fraction are cancerous, with papillary carcinoma being the most commonly observed histologic type [1,2,3]. Therefore, ensuring an accurate diagnosis of thyroid nodules is crucial to avoid unnecessary surgeries and enable timely treatment in cases of malignancy. Ultrasound is the most commonly employed technique for detecting and diagnosing thyroid diseases, particularly thyroid nodules, as it enables qualitative assessment of these lesions. However, the reliability of ultrasounds remains a subject of debate, as it relies on the sensitivity of the equipment and the operator’s experience with the specific method [4,5,6]. Fine needle aspiration cytology (FNAC) is the primary diagnostic investigation for thyroid nodules. It involves collecting cells from the nodule and examining them under a microscope. Several studies have compared the diagnostic accuracy of FNAC and histopathology in diagnosing thyroid nodules. Overall, FNAC has reported sensitivity and specificity rates of 79–95% and 72–100%, respectively, in detecting thyroid malignancy. Conversely, histopathology has reported sensitivity and specificity rates of 80–100% and 97–99%, respectively [7]. FNAC is a less invasive alternative to histopathology, as it involves only the insertion of a needle into the thyroid [7]. However, FNAC does have certain limitations. It may not yield optimal results for patients with lesions smaller than 1 cm, thyroiditis, and follicular neoplasms or in cases where malignancy cannot be definitively determined. In contrast, histopathologic diagnosis is a more invasive and costly procedure, necessitating the removal of either the entire nodule or the affected portion of the thyroid gland. Unfortunately, a notable percentage of thyroid nodules may undergo unnecessary surgical removal to facilitate histopathologic diagnoses. Furthermore, histopathology is typically conducted post-surgery and tends to be more time-consuming than FNAC. Therefore, the aim of our study was to compare the diagnostic accuracy of FNAC and ultrasound in diagnosing thyroid nodules, using histopathology as the reference standard.

## 2. Materials and Methods

A retrospective and transversal study was conducted on patients admitted to the Endocrine and ENT Surgery Department of Hôtel-Dieu de France Hospital in Lebanon between January 2017 and May 2022. The study included patients first subjected to a thyroid ultrasound, followed or not by an FNAC, who then had thyroid gland surgery. Patients without a preoperative ultrasound report or with missing histopathology results were excluded. Ultrasound of the thyroid gland is mandatory before thyroidectomy. Ethical approval was obtained from the medical research center, and data were extracted from the institutional database for analysis. Histopathological findings were compared to preoperative ultrasound and thyroid needle aspiration reports. In this study, fine needle aspiration was conducted either under ultrasound guidance or by palpation without the need for imaging. In instances of multiple nodules, even if one nodule was determined to be malignant and the others benign, the case was classified as malignant based on cytology to ensure accurate classification.

The EU-TIRADS classification for thyroid ultrasounds was adopted in this study. This imaging system assigns a predictive score ranging from 1 to 5, with the following categories: EU-TIRADS 1: No nodule; EU-TIRADS 2: Benign; EU-TIRADS 3: Slightly suspicious; EU-TIRADS 4: Moderately suspicious; and EU-TIRADS 5: Highly suspicious [8].

If FNA was performed on the thyroid nodules, the cytological diagnosis was determined based on the international cytological classification of Bethesda. Therefore, the cytopathology reports were categorized as follows: Bethesda I (unsatisfactory material), Bethesda II (benign), Bethesda III (atypical/follicular lesion of undetermined significance), Bethesda IV (suspected follicular neoplasia), Bethesda V (suspected malignancy), and Bethesda VI (malignancy). Subsequently, the ratios obtained from the ultrasound, cytology (when available), and histopathology were compared, and the data were subjected to analysis [9].

### Statistical Analysis

All the collected data were compiled in MS Excel for analysis. Sensitivity, specificity, positive predictive value (PPV), negative predictive value (NPV), false positive rate, false negative rate, and overall accuracy were calculated for each diagnostic method. The risk of malignancy associated with each EU-TIRADS category and Bethesda category was determined by comparing them with the reference histopathology reports. Statistical analyses were conducted using IBM SPSS 20.0 software. Pearson’s correlation coefficient (r-test) was used to analyze the correlation among the reported results of these three diagnostic methods. A probability value of *p* ≤ 0.05 was considered indicative of statistical significance. The definitions of false positives (FP), true positives (TP), false negatives (FN), and true negatives (TN) are as follows: A true negative refers to cases where the histological finding indicates hyperplasia or inflammation and falls under the Bethesda II category. A false negative occurs when a neoplasm (malignant or benign) is present but categorized as Bethesda II. A true positive corresponds to malignant neoplasms categorized under Bethesda V and VI or malignant/benign follicular neoplasms categorized under Bethesda III and IV. Finally, a false positive is indicated when hyperplastic/inflammatory processes are categorized as Bethesda V and VI or benign non-neoplastic diseases are categorized as Bethesda III and IV [8,9,10].

## 3. Results

### 3.1. Patient Characteristics

A total of 357 patients underwent total thyroidectomy or lobo-isthmectomy during the specified period, of which 344 patients were enrolled in this study. The inclusion criteria were all patients with thyroidectomy done in our hospital between 2017 and 2022, with an ultrasound examination done before the surgery and a histopathology report available for each one. Among the study population of 344 patients, 259 (75.30%) were women, and 85 (24.70%) were men, resulting in a female-to-male ratio of 3:1. The mean age of the study population was 49.02 ± 15.17 years, ranging from 13 to 91 years. Following surgery, thyroid tumors in 61.33% (211 cases) of the patients were identified as benign, while the remaining 38.67% (133 cases) were diagnosed as malignant. This translates to a ratio of 1:1.6 between malignant and benign lesions. Furthermore, higher malignancy rates were observed in males than in females. Among the 133 patients with malignant histology, 42 out of 85 males (49.41%) and 91 out of 259 females (35.13%) were affected.

### 3.2. Paraclinical Data

The distribution of EU-TIRADS classification categories was as follows: 3 patients with EU-TIRADS 2 (0.88%), 202 patients with EU-TIRADS 3 (58.72%), 117 patients with EU-TIRADS 4 (34%), and 22 patients with EU-TIRADS 5 (6.40%). A very limited number of patients fell into the EU-TIRADS category 2, and, therefore, they were excluded from the statistical analyses. The risk of malignancy was calculated for each category based on the histopathological findings, revealing an ascending trend from EU-TIRADS categories 3 to 5. Table 1 presents the ultimate histopathological diagnosis in the various sonographic categories as per the EU-TIRADS classification.

Table 2 displays the preoperative EU-TIRADS score alongside the histopathological findings. By grouping categories 4 and 5 of EU-TIRADS classification (moderately and highly suspicious lesions) together and considering them as potentially malignant sonographic divisions while classifying EU-TIRADS 3 as a probable benign ultrasound category, the sensitivity, specificity, positive predictive value (PPV), and negative predictive value (NPV) were determined to be 68.18%, 76.55%, 64.74%, and 79.20%, respectively. The overall accuracy of ultrasounds was calculated to be 73.31%.

Out of the 344 patients who underwent thyroidectomy, fine needle aspiration (FNA) was performed on 201 individuals. Among these 201 FNA cases, 45 (22.39%) were determined to be benign based on cytology, 44 (21.89%) were categorized as atypical or follicular lesions of undetermined significance, 38 (18.90%) were interpreted as follicular neoplasms or suspicious lesions, 63 (31.34%) were considered suspicious for malignancy, and 5 (2.49%) were confirmed as malignant. There were six additional cases (2.99%) that did not yield a satisfactory cytological evaluation. In the current study, most results fell into Bethesda category V (31.34%) rather than category II, as expected by the distribution of the Bethesda system, followed by types II and III, with 22.39% and 21.89%, respectively. This comparison is presented in Table 3, along with the differences between the risk of malignancy found in our study and the rates predicted by the Bethesda classification. The malignancy rates observed in our study for categories I to VI, based on postoperative histopathological results, were 16.67%, 20%, 54.54%, 34.21%, 88.89%, and 100%, respectively.

### 3.3. Anatomo-Pathological Study

Regarding the final histopathological diagnosis of the operated thyroid glands, 211 patients were found to have benign lesions. Among them, 114 (54.03%) were diagnosed with colloid goiter, 33 (15.64%) with chronic lymphocytic thyroiditis, 30 (14.22%) with a follicular neoplasm, 22 (10.43%) with a colloid nodule, 9 (4.27%) with NIFTP (non-invasive follicular thyroid neoplasm with papillary-like features), and the remaining 3 patients (1.42%) had dystrophic thyroid parenchyma. Additionally, 133 patients had malignant lesions. The majority of those lesions were papillary carcinomas found in 120 (90.23%) patients. This was followed by follicular carcinoma in seven (5.27%), oncocytic carcinoma in four (3%), and medullary carcinoma and Hurthle cell carcinoma in one patient each (0.75%, 0.75%).

### 3.4. Correlation

#### 3.4.1. Correlation Analysis between Ultrasound and FNA

Table 4 shows the correlation between two diagnostic methods: ultrasound and CPAF. Out of the 202 patients who were deemed less suspicious based on ultrasound findings, only 80 patients underwent additional fine needle aspiration biopsy (FNAB) prior to surgery. According to the result of the Bethesda classification, 5% were categorized as unsatisfactory, 46.25% as benign, 23.75% as atypia of undetermined significance, 13.75% as follicular neoplasms, and 11.25% as suspected malignant. None of the cases in this group were diagnosed as malignant. On the other hand, among the 117 patients classified as moderately suspicious and the 22 patients classified as highly suspicious based on EU-TIRADS classification, only 103 patients underwent FNAB in the first group. In this subgroup, the results showed that 1.94% were unsatisfactory, 7.76% were benign, 21.36% were atypia of undetermined significance, 23.30% were follicular neoplasms, 42.72% were suspected as malignant, and 2.92% were diagnosed as malignant. For the 18 patients in the highly suspicious group who underwent FNAB, the results indicated that 16.67% were classified as atypia of undetermined significance, 16.67% as follicular neoplasms, 55.55% as suspected malignant, and 11.11% as malignant. Notably, no results fell into the first two categories of the Bethesda classification. The correlation coefficient (Pearson’s r) between ultrasound findings and fine needle aspiration biopsy results was calculated to be 0.512, which was found to be statistically significant (*p* < 0.0000001).

#### 3.4.2. Evaluation of the Correlation between Ultrasound and Histological Results

Table 5 shows the correlation between ultrasound and pathology as diagnostic methods. Based on the ultrasound findings, 202 patients were classified as having low suspicion, 117 as having intermediate suspicion, and 22 as having high suspicion. Subsequent pathological examination revealed malignancy in 20.80% of patients in the low suspicion group, 60.70% in the medium suspicion group, and 86.36% in the high suspicion group. The correlation coefficient (Pearson’s r) between these two methods was calculated to be 0.408, indicating a statistically significant relationship (*p* < 0.0000001).

#### 3.4.3. Evaluation of the Correlation between FNAB and Histological Results

According to the presented tables, all thyroid glands that exhibited malignancy on cytology were confirmed to have malignant histology postoperatively (Table 6). Out of the 45 thyroid glands with benign cytology, 9 (20%) were found to be histologically malignant. The decision to perform surgery for this group was primarily based on factors such as rapid nodule growth and the development of symptomatic large-sized nodules. In Bethesda classes III and IV, 24 out of 44 cases (54.54%) and 13 out of 38 cases (34.2%), respectively, were histologically malignant. In the Bethesda V group, 56 out of 63 cases (88.9%) were found to be malignant. The sensitivity and specificity of fine-needle aspiration cytology were determined to be 89.31% and 48.44%, respectively. The false positive rate was 51.56%, while the false negative rate was 10.68%. The positive predictive value was 78%, and the negative predictive value was 68.89%. The accuracy of fine needle aspiration cytology in distinguishing between benign and malignant thyroid lesions was calculated as 75.89%. As previously mentioned, among the 344 cases, 211 were confirmed as benign through pathology, with 54 cases demonstrating a discrepancy between fine needle aspiration and pathology findings (Table 7). Based on the collected data, Pearson’s r correlation coefficient between these two methods was determined to be 0.304, which is statistically significant (*p* < 0.00001).

#### 3.4.4. Evaluation of the Correlation between Ultrasound, Cytology and Pathology

Table 4 shows the correlation between the three different diagnostic methods: ultrasound, CPFA, and histology. First, a group of 202 patients was initially diagnosed with low sonographic suspicion. Surprisingly, 80 patients (39.60%) underwent FNAB before surgery. Next, a group of 117 patients was diagnosed as moderately suspicious according to the EU-TIRADS classification. Thus, out of these, 103 patients (88%) underwent additional cytological examination before surgery. Finally, 22 patients were diagnosed as highly suspicious of malignancy based on ultrasound findings. Consequently, approximately 82% (18/22) of them underwent FNAB followed by surgery. The Pearson test yielded a *p* value of 0.00001, indicating a statistically significant correlation coefficient of 0.423 among these three diagnostic methods (*p* < 0.0001).

## 4. Discussion

The primary objective of our monocentric study was to investigate the correlation between cytology and histology, as well as between ultrasound and histology. Additionally, we aimed to evaluate whether the incidence of malignancy in each EU-TIRADS class aligned with the estimations provided by experts from the European Thyroid Association (ETA). Upon analyzing our findings, we observed a correlation coefficient of 0.304 for the cytology-histology comparison and 0.408 for the ultrasound-histology comparison.

The distribution of patients with thyroid dysmorphism revealed a peak occurrence in the 4th, 5th, and 6th decades of life. These findings align with previous studies [11,12] that have also reported a higher prevalence of thyroid abnormalities in older age groups. In contrast with the existing literature [13,14], our study revealed a higher rate of malignancy in men compared to women.

In our department, we have implemented the new EU-TIRADS classification developed by the ETA to stratify ultrasound results. This classification system assesses the presence of five lesion prototypes of equal significance and assigns nodules to one of the five categories based on the number of suspicious characteristics [15]. This system is known for its practicality and accuracy. Our findings revealed a malignancy prevalence of 33.33% in EU-TIRADS class 2, 20.80% in class 3, 60.68% in class 4, and 86.36% in class 5. It is interesting to note that these results differ significantly from the estimates provided by ETA experts. According to their estimates, the malignancy rates are close to zero in EU-TIRADS class 2, ranging from 2% to 4% in class 3, 6% to 17% in class 4, and 26% to 87% in class 5 [15]. The prevalence of malignancy in EU-TIRADS classes 3 and 4 was found to be 10 times higher than the estimates provided by ETA experts. This disparity can be attributed to the presence of small malignant nodules in close proximity to nodules that were initially interpreted as less suspicious on ultrasound. It is important to note that many patients in our study had undergone ultrasound examinations elsewhere, which prevented us from verifying the quality and accuracy of those examinations.

During the prospective validation of the TIRADS classification conducted by Horvath et al. [16] on 502 resected thyroid nodules, higher sensitivity, specificity, positive predictive value (PPV), and negative predictive value (NPV) (99.6%, 74.4%, 82.1%, and 99.4%, respectively) were reported compared to our study. However, our findings are consistent with those of [17], who demonstrated sensitivity and specificity values of 75% and 77.91% and 74% and 83%, respectively, for ultrasound in suggesting malignant disease. In our study, we detected a weak positive correlation (r = 0.408) between this non-invasive diagnostic method and the final postoperative diagnosis despite a strong positive correlation being found in the literature [18]. Therefore, ultrasound has the advantage of providing a comprehensive evaluation of the entire thyroid gland rather than just the dominant nodule. However, it is important to note that ultrasound findings alone do not provide definitive evidence of malignancy, as no single feature is pathognomonic. Consequently, ultrasound of the thyroid should be considered a complementary tool to fine needle aspiration rather than an alternative, assisting in the comprehensive evaluation of thyroid nodules.

Regarding cytopathology, only 201 patients underwent FNAB following ultrasound evaluation but prior to surgery. The remaining 143 patients in our population opted for direct thyroidectomy despite 3 patients being classified as EU-TIRADS 2 and 122 as EU-TIRADS 3. The primary reasons for this decision among these 125 patients were symptoms of local compression, potential future malignancy risk, or aesthetic concerns. However, the remaining 18 patients, with 14 classified as EU-TIRADS category 4 and 4 as EU-TIRADS category 5, chose direct surgical treatment over FNAB to minimize follow-up procedures.

This allowed us to identify a strong positive correlation (r = 0.512) between the non-invasive diagnostic method and the minimally invasive approach, which aligns with the correlation recognized in the existing literature [18].

In contrast to our incidence distribution, Machala et al. [19] reported different proportions in each Bethesda category. They found 12.52% of their population in Bethesda category I (compared to 3% in our study), 58.95% in category II (compared to 22.35% in our study), 6% in category III (compared to 21.90% in our study), 7.29% in category IV (compared to 18.90% in our study), 11.33% in category V (compared to 31.35% in our study), and 3.88% in category VI (compared to 2.50% in our study). Consequently, their distribution was in agreement with the incidence predicted by Bethesda [20], which suggests that the highest incidence is in category II rather than in the fifth category as found in our population. The lower incidence of benign cytology found in our study can be attributed to the fact that our study is a retrospective evaluation of a highly selected group of patients who underwent thyroidectomy, possibly already presenting an increased risk for malignancy and excluding other patients who underwent FNAB during the last five years in the hospital. Furthermore, when analyzing each category, the malignancy rate in our study was generally higher than the rates predicted by the Bethesda system [21], further supporting our theory.

Limitations of FNAB include the possibility of false negative and false positive results. In a comparative study conducted by Bloch et al. [22], the accuracy of FNA was reported to be 91.6% when compared to histological results. Similarly, Handa et al. [23] found that FNA had a sensitivity of 97%, specificity of 100%, positive predictive value (PPV) of 96%, and negative predictive value (NPV) of 100%. Another study by Mundasad et al. [24] demonstrated a sensitivity of 52.6%, specificity of 86.6%, and accuracy of 79.1% for FNA in determining thyroid gland malignancy. Overall, FNA exhibits excellent sensitivity and specificity, with reported ranges of 65–98% (mean 83%) and 72–100% (mean 92%), respectively [25]. In addition, the PPV of FNA varies between 50% and 96%, with an average of 75% [26], while the NPV ranges from 57% to 100% [24,27]. However, in our study, FNAB showed a sensitivity of 89.30%, consistent with the majority of studies, but had a low specificity of 48.45%. The positive and negative predictive values were 78% and 69%, respectively, and the overall diagnostic performance of CPAF was 73.65%. These findings suggest that FNA is effective in detecting malignancy with high sensitivity but may be less reliable in identifying benign nodules.

The rate of false negatives in FNA can vary significantly, ranging from 0.7% to 13%, and can be as high as 30% when considering large solid and cystic nodules [28]. Our results were, therefore, in agreement with the literature, with a rate of false negatives of 10.7%. However, what stood out in our study was the remarkably high false positive rate of 51.55%, compared to the reported rates of 0% to 7% in the literature [26]. This suggests that in our institution, we perform a high number of cyst punctures, which can be excessive in certain cases and leads to unnecessary procedures in particular situations. This overdiagnosis of thyroid cancers could explain the higher malignancy rate observed in our study.

In our study, we observed a weak positive correlation (r = 0.304) between this minimally invasive diagnostic method and the final postoperative diagnosis, whereas the literature suggests a strong positive correlation [18].

Regarding benign nodules, the majority (76.77%) obtained an EU-TIRADS score of less than or equal to 3, indicating a low suspicion of malignancy. However, a notable proportion (23.23%) received an EU-TIRADS score of 4 or 5, which would have led to unnecessary thyroid cytopunctures for a benign histological diagnosis. It is important to note that the primary objective of thyroid ultrasound, regardless of the scoring system used, is to minimize or even eliminate unnecessary thyroid cytopunctures.

In most Bethesda categories, discordant cases with histopathological findings were observed, as shown in Table 7. It should be noted that among the 63 patients classified as Bethesda V, 56 had malignant histopathology, of whom 7 were in the EU-TIRADS 4 category with nodules <15 mm and 2 were in the EU-TIRADS 5 category with nodules <10 mm. Based on these findings, we were able to determine that among the 201 patients who underwent FNAB, 18 patients (1 in B II, 6 in B III, 2 in B IV, and 9 in B V) underwent a cytopuncture when it was not indicated by the ETA guidelines [15].

However, in these specific cases, FNAB proved to be necessary and provided valuable guidance in suspecting malignant pathologies and determining the need for surgical intervention. Thus, the combination of both examinations holds significance and demonstrates a high positive predictive value, particularly in cases of Bethesda Group IV.

Finally, our study revealed a positive correlation among the three diagnostic methods examined. According to the Pearson test, the correlation coefficient (r) was determined to be 0.423, indicating a weak positive correlation between thyroid ultrasound, fine needle aspiration cytology, and histopathology. It is worth noting that the literature has reported a strong correlation among these three modalities [18].

This study has several limitations that should be acknowledged. Firstly, it is important to note that this study is retrospective in nature and was conducted at a single tertiary center. Consequently, the generalizability of the findings may be limited, as studies encompassing multiple centers often exhibit greater heterogeneity in patient populations and nodule types, yielding potentially different outcomes. Secondly, a portion of the patients included in this study were referred from different institutions and already had ultrasound and fine needle aspiration reports. It is possible that these institutions may have varying levels of experience and expertise in diagnosing thyroid nodules, which could introduce variability in the data. Thirdly, the indication for surgery and the inclusion of non-operated patients were not considered in our analysis. These omissions may have contributed to the higher rates of thyroid cancer observed in our study. To validate the results of our study, it is recommended that future research be conducted, particularly in other tertiary centers across Lebanon.

## 5. Conclusions

This study establishes a correlation between ultrasound characteristics of thyroid nodules with cytology and histopathology. Based on the findings, it can be concluded that fine-needle aspiration biopsy demonstrated higher sensitivity than EU-TIRADS in detecting thyroid cancer, although it was less specific. Both methods showed comparable overall accuracy. The diagnostic approach should be tailored to each patient, considering their clinical and radiologic findings. Fine-needle aspiration cytology should be performed in all suspicious cases, reserving surgical evaluation for those with a substantial risk of malignancy.

## Figures and Tables

**Table 1 jpm-13-01197-t001:** Comparison between the proportions of malignancies in our study and those predicted by the EU-TIRADS classification.

EU-TIRADS Score	Number of Patients %	Histopathologic Findings *n*		Risk of Malignancy (%)	Risk of Malignancy Predicted by the EU-TIRADS Classification (%)
		**Malignant**	**Benign**		
2	3 (0.88%)	1	2	33.33	≅0
3	202 (58.72%)	42	160	20.8	2–4
4	117 (34%)	71	46	60.68	6–17
5	22 (6.40%)	19	3	86.36	26–87
Total	344 (100%)	133	211	38.67	

**Table 2 jpm-13-01197-t002:** Comparison of TIRADS and risk of malignancy.

Preoperative Investigation	Histopathology		
Ultrasound	Malignant *n* (%)	Benign *n* (%)	Total *n* (%)
Eu-Tirads 3	42 (20.79%)	160 (79.21%)	202 (59.23%)
Eu-Tirads 4, 5	90 (64.74%)	49 (35.26%)	139 (40.77%)
Total	132 (38.70%)	209 (61.30%)	341 (100%)

**Table 3 jpm-13-01197-t003:** Cytological distribution and proportion of malignancy according to the Bethesda classification.

Cytology According to the Bethesda Classification	Expected Incidence by Bethesda (%)	Incidence in Our Study (%)	Expected Malignancy by Bethesda (%) [9]	Malignancy in Our Study (%)
Category 1—Non-diagnostic	5–11%	2.99%	5–10%	16.67%
Category 2—Benign	55–74%	22.39%	0–3%	20%
Category 3—Atypia of undetermined significance or follicular lesion of undetermined significance	5–15%	21.89%	6–18%	54.54%
Category 4—Suspicious for follicular neoplasm	2–25%	18.90%	10–40%	34.21%
Category 5—Suspicious for malignant	1–6%	31.34%	45–60%	88.89%
Category 6—Malignant	2–5%	2.49%	94–96%	100%

**Table 4 jpm-13-01197-t004:** Correlation between ultrasound and cytological and histological diagnostic methods.

		Histology	
Ultrasound	Number	Malignant (Number [%])	Benign (Number [%])
Low risk (EU-TR3)	**202**	Papillary Carcinoma (38 [18.80%])	Multinodular Goiter (94 [46.52%])
		Follicular Carcinoma (2 [1%])	Lymphocytic Thyroiditis (30 [14.85%])
		Oncocytic carcinoma (1 [0.5%])	Follicular Neoplasm (18 [8.90%])
		Medullary Carcinoma (1 [0.5%])	Solitary Nodule (13 [6.43%])
		Hurthle cell Carcinoma (0 [0%])	NIFTP (3 [1.5%])
			Dystrophic Thyroid Parenchyma (2 [1%])
	Total	**42 (20.80%)**	**160 (79.20%)**
Intermediate risk (EU-TR4)	**117**	Papillary Carcinoma (65 [55.55%])	Multinodular Goiter (16 [13.68%])
		Follicular Carcinoma (4 [3.43%])	Lymphocytic Thyroiditis (3 [2.57%])
		Oncocytic carcinoma (1 [0.85%])	Follicular Neoplasm (12 [10.25%])
		Medullary Carcinoma (0 [0%])	Solitary Nodule (9 [7.70%])
		Hurthle cell Carcinoma (1 [0.85%])	NIFTP (5 [4.27%])
			Dystrophic Thyroid Parenchyma (1 [0.85%])
	Total	**71 (60.68%)**	**46 (39.32%)**
High risk (EU-TR5)	**22**	Papillary Carcinoma (16 [72.72%])	Multinodular Goiter (3 [13.64%])
		Follicular Carcinoma (1 [4.54%])	Lymphocytic Thyroiditis (0 [0%])
		Oncocytic carcinoma (2 [9.1%])	Follicular Neoplasm (0 [0%])
		Medullary Carcinoma (0 [0%])	Solitary Nodule (0 [0%])
		Hurthle cell Carcinoma (0 [0%])	NIFTP (0 [0%])
			Dystrophic Thyroid Parenchyma (0 [0%])
	Total	**19 (86.36%)**	**3 (13.64%)**

**Table 5 jpm-13-01197-t005:** Correlation between ultrasound and histological diagnostic methods.

		Histology	
FNA	Number	Malignant *n* (%)	Benign *n* (%)
**B1**: Non-diagnostic	6	1 (16.67%)	5 (83.33%)
**B2**: Benign	45	9 (20%)	36 (80%)
**B3**: Atypia of undetermined significance or follicular lesion of undetermined significance	44	24 (54.54%)	20 (45.46%)
**B4**: Suspicious for follicular neoplasm	38	13 (34.21%)	25 (65.79%)
**B5**: Suspicious for malignancy	63	56 (88.89%)	7 (11.11%)
**B6**: Malignant	5	5 (100%)	0 (0%)

**Table 6 jpm-13-01197-t006:** Correlation between cytological and histological diagnostic methods.

		Histology	
FNA	Number	Malignant *n* (%)	Benign *n* (%)
**B1**: Non-diagnostic	6	1 (16.67%)	5 (83.33%)
**B2**: Benign	45	9 (20%)	36 (80%)
**B3**: Atypia of undetermined significance or follicular lesion of undetermined significance	44	24 (54.54%)	20 (45.46%)
**B4**: Suspicious for follicular neoplasm	38	13 (34.21%)	25 (65.79%)
**B5**: Suspicious for malignancy	63	56 (88.89%)	7 (11.11%)
**B6**: Malignant	5	5 (100%)	0 (0%)

**Table 7 jpm-13-01197-t007:** Table showing details of the discordant cases.

Cytology According to the Bethesda Classification	Number of Cases			Histopathology			EU-TIRADS and Size of Nodule (*n*)
		**Benign (*n*)**	**Papillary Carcinoma (*n*)**	**Follicular Carcinoma (*n*)**	**Oncocytic Carcinoma (*n*)**	**Other Carcinomas (*n*)**	
I	1 of 6 (16.67%)	-	0	1	0	0	1 EU-3 (>20 mm)
II	9 of 45 (20%)	-	7	1	1	0	7 EU-3 (tous > 20 mm) 2 EU-4 (dont 1 < 15 mm)
III	24 of 44 (54.54%)	-	21	2	0	1	8 EU-3 (dont 1 < 20 mm) 13 EU-4 (dont 5 < 15 mm) 3 EU-5 (tous > 10 mm)
IV	13 of 38 (34.21%)	-	9	1	3	0	3 EU-3 (tous > 20 mm) 8 EU-4 (dont 2 < 15 mm) 2 EU-5 (tous > 10 mm)
V	7 of 63 (11.11%)	7	-	-	-	-	3 EU-3 (tous > 20 mm) 4 EU-4 (tous > 15 mm)
VI	0 of 5 (0%)	0	-	-	-	-	-
Total *n* (%)	54 of 201 (26.86%)	7 (3.49%)	37 (18.40%)	5 (2.49%)	4 (1.99%)	1 (0.49%)	
% Distribution of discordant cases out of total 54 cases	100%	12.96%	68.53%	9.26%	7.40%	1.85%	

## Data Availability

Data available on request due to ethical restrictions. The data presented in this study are available on request from the corresponding author.

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
