# Peer review of "Comparison between Fine Needle Aspiration Cytology with Histopathology in the Diagnosis of Thyroid Nodules"

_jpm, 2023, doi:10.3390/jpm13081197_

Round 1

Reviewer 1 Report

Osseis et al. compare the diagnostic accuracy of thyroid nodule FNAC and ultrasound using histopathology. The paper is correctly organized, study is retrospective, topic is geographically and locally focused. But, we do not hear anything new about that clinically very important topic. I am just thinking about belonging of presented paper to  Novel Challenges and Therapeutic Strategies for Ear, Nose and Throat (ENT) Diseases.

Discussion is very thin, some very important refences are missing. The conclusions are too general. 

------------------------------------------------------------------------------

1. What is the main question addressed by the research?

 Osseis et al. calculated sensitivity, specificity, positive and negative predictive values, false positive and negative rates, and overall accuracy for thyroid ultrasound and FNAC regarding more precise assessment of thyroid nodules (TN). 

 2. Do you consider the topic original or relevant in the field? Does it address a specific gap in the field?

 The topic is actual in last decade and much effort is given to analyze benign or malignant nature in more precise manner. Many studies have already analyzed the accuracy of US and FNAC. But, presented study is regionally specific and could address a specific gap in such geographic area.

 3. What does it add to the subject area compared with other published material?

Both FNAC and thyroid ultrasound demonstrate moderate diagnostic accuracy regarding TN. Expectedly, FNAC has had higher sensitivity, opposite to thyroid ultrasound that demonstrated higher specificity. That is the answer why presented techniques are usually used together.

4. What specific improvements should the authors consider regarding the methodology? What further controls should be considered?

There is no place for any improvement in methodology, but in addition of some molecular or biochemical marker to improve accuracy in malignant TN diagnostics.

 5. Are the conclusions consistent with the evidence and arguments presented and do they address the main question posed?

Conclusions are consistent with findings and they address the main question posed.

6. Are the references appropriate?The references are appropriate.

7. Please include any additional comments on the tables and figures. The figures and tables are correct.   Presented manuscript is well organized and correctly presented. Editor must make a decision whether this manuscript presented something new in assessment of thyroid nodule. I stress again that the main advantage of paper is the presentation of region-specific analysis. 

Thanks for collaboration. 

Moderate English polishing is obliged.

Author Response

Replying to reviewer 1 :

We thank the reviewer for his valuable comments after reviewing our article. Please find below our response to these comments. Modifications were made according the reviewer comments

Extensive editing of English language was accomplished by the proofreadingServices.com . (Certificate attached below )

Quality of English Language 

(x) Minor editing of English language required ( ) English language fine. No issues detected 

Extensive editing of English language was accomplished by the proofreadingServices.com . (Certificate attached below )

Osseis et al. compare the diagnostic accuracy of thyroid nodule FNAC and ultrasound using  histopathology. The paper is correctly organized, study is retrospective, topic is  geographically and locally focused. But, we do not hear anything new about that clinically  very important topic. I am just thinking about belonging of presented paper to  Novel Challenges and Therapeutic Strategies for Ear, Nose and Throat (ENT) Diseases.  Discussion is very thin, some very important refences are missing. The conclusions are too general. 

We are grateful for your time spent reviewing our paper. Like you already mentioned, this is a retrospective study evaluating the correlation of fine needle aspiration cytology with ultrasound and final histology in the diagnosis of thyroïde nodules. It would be very interesting if you can help us citing these missing references to add it to our article. We will try to be more precise in our discussion and conclusion.

 1. What is the main question addressed by the research? 

Osseis et al. calculated sensitivity, specificity, positive and negative predictive values, false positive and negative rates, and overall accuracy for thyroid ultrasound and FNAC regarding more precise assessment of thyroid nodules (TN). 

Thank you. Yes the main question was to calculate the sensitivity, specificity , PPV and NPV with the correlation between Ultrasound, FNAC and final histology in thyroïd nodules. 

2. Do you consider the topic original or relevant in the field? Does it address a specific gap in the field? 

The topic is actual in last decade and much effort is given to analyze benign or malignant nature in more precise manner. Many studies have already analyzed the accuracy of US and FNAC. But, presented study is regionally specific and could address a specific gap in such geographic area. 

Thank you for your response. Indeed this is the first study evaluating this correlation in a tertiary center in the middle East Region.

3. What does it add to the subject area compared with other published material? 

Both FNAC and thyroid ultrasound demonstrate moderate diagnostic accuracy regarding TN. Expectedly, FNAC has had higher sensitivity, opposite to thyroid ultrasound that demonstrated higher specificity. That is the answer why presented techniques are usually used together. 

Thank you.

4. What specific improvements should the authors consider regarding the methodology? What further controls should be considered? 

There is no place for any improvement in methodology, but in addition of some molecular or biochemical marker to improve accuracy in malignant TN diagnostics. 

Thank you. Unfortunately our database is missing some molecular markers because measurements of these markers preoperatively is not mandatory.

5. Are the conclusions consistent with the evidence and arguments presented and do they address the main question posed? 

Conclusions are consistent with findings and they address the main question posed. 

Thank you.

6. Are the references appropriate?The references are appropriate.

Thank you. Some old references were changed.

7. Please include any additional comments on the tables and figures. 

The figures and tables are correct. Presented manuscript is well organized and correctly presented. Editor must make a decision whether this manuscript presented something new in assessment of thyroid nodule. I stress again that the main advantage of paper is the presentation of region-specific analysis. 

Thank you Again. We improved the presentation of the tables.

Comments on the Quality of English Language 

Moderate English polishing is obliged. 

Extensive editing of English language was accomplished by the proofreadingServices.com . (Certificate attached below )

Reviewer 2 Report

I carefully read the article sent. The study carried out by the authors despite being a retrospective one is very well done. The obtained results are important in daily practice. The quality of the presentation is very good.

minor changes

Author Response

Replying to Reviewer 2 :

We Thank the reviewer for his valuable comments after reviewing our article. Please find below our response to these comments. Modifications were made according to the reviewer comments.

Extensive english editing was accomplished by the proofreadingServices.com (Certificate attached below)

Reviewer 3 Report

Michael Osseis et al have made a very well-organized comparison between FNAC and the histopathological diagnosis of the thyroid nodules.

My suggestions are as follows:

ABSTRACT

The abstract is well written, emphasizing the most important findings.

Introduction:

Line 55-65: There is no bibliography reference to these statements.

Materials and Methods

The study is a retrospective and transversal study, not a retrospective review (line 67).

Please use a bibliographic reference for the TIRADS and Bethesda Classification (and the edition used).

Please re-write the inclusion and exclusion criteria of the patients into your study in order to be clear by which criteria were they selected.

Results

For Table 1 – make sure that you define in the legend row the unit of measurement (%, n) for each variable that you use.

The same for Table 2 – you can write for example Malignant (n). In the first row, first column, there is a “164” which I think that is a mistake.

Table 3 – as I recommended before, please specify the edition of the Bethesda Classification that you used, because you used the percentages given by Bethesda in your study – and in time this percentages can change.

Please apply my comments to every table.

Discussion

The discussion part is well written, with adequate comparison from the literature.

Conclusion:

The conclusion is supported by the results and conclusion.

Bibliography:

Some of your bibliographic references are too old (1995, 1993). Please update them to more relevant studies from our days, because you have to compare your results with data from our days, not from 1992-1995 (30 years ago).

Author Response

Replying to Reviewer 3 :

We Thank the reviewer for his valuable comments after reviewing our article. Please find below our response to these comments. Modifications were made according to the reviewer comments.

Extensive english editing was accomplished by the proofreadingServices.com (Certificate attached below)

ABSTRACT

The abstract is well written, emphasizing the most important findings. 

Thank You

Introduction:

Line 55-65: There is no bibliography reference to these statements. 

Thank you. A Reference were added to the statement. “Jasim S, Dean DS, Gharib H. Fine-Needle Aspiration of the Thyroid Gland. In: Feingold KR, Anawalt B, Blackman MR, Boyce A, Chrousos G, Corpas E, et al., editors. Endotext. South Dartmouth (MA): MDText.com, Inc.; 2000. »

Materials and Methods

The study is a retrospective and transversal study, not a retrospective review (line 67).

Thank you for your comment. “A retrospective and transversal study was conducted «  was added in line 69.

Please use a bibliographic reference for the TIRADS and Bethesda Classification (and the edition used).

Thank you.

References were added

Russ G, Bonnema SJ, Erdogan MF, Durante C, Ngu R, Leenhardt L. European Thyroid Association Guidelines for Ultrasound Ma-lignancy Risk Stratification of Thyroid Nodules in Adults: The EU-TIRADS. Eur Thyroid J. 2017 Sep;6(5):225–37.

Cibas ES, Ali SZ. The 2017 Bethesda System for Reporting Thyroid Cytopathology. Thyroid. 2017 Nov;27(11):1341–6.

Please re-write the inclusion and exclusion criteria of the patients into your study in order to be clear by which criteria were they selected. 

Sure. Some improvements were made to our text by adding this (Line 118-120):

“The inclusion criteria were all patients with thyroidectomy done in our hospital be-tween 2017 and 2022, with an ultrasound examination done before the surgery and a histopathology report available for each one. “

Results

For Table 1 – make sure that you define in the legend row the unit of measurement (%, n) for each variable that you use. 

Thank you. We added the measurement units in all tables

The same for Table 2 – you can write for example Malignant (n). In the first row, first column, there is a “164” which I think that is a mistake. 

Thank you. We added the measurement units in all tables

Table 3 – as I recommended before, please specify the edition of the Bethesda Classification that you used, because you used the percentages given by Bethesda in your study – and in time this percentages can change. 

The reference concerning the Bethesda was added to the table 3.

Please apply my comments to every table.

Done. Thank you

Discussion

The discussion part is well written, with adequate comparison from the literature. 

 Thank you.

Conclusion:

The conclusion is supported by the results and conclusion. 

Round 2

Reviewer 1 Report

Accept the paper in revised form.

Reviewer 3 Report

The authors have revised their manuscript in concordance with my suggestions.